# Bioactive Compounds of *Nigella Sativa* Essential Oil as Antibacterial Agents against *Chlamydia Trachomatis* D

**DOI:** 10.3390/microorganisms7090370

**Published:** 2019-09-19

**Authors:** Tímea Mosolygó, Ahmad Mouwakeh, Munira Hussein Ali, Annamária Kincses, Csilla Mohácsi-Farkas, Gabriella Kiskó, Gabriella Spengler

**Affiliations:** 1Department of Medical Microbiology and Immunobiology, Faculty of Medicine, University of Szeged, 6720 Szeged, Hungary; munirahusseinali@gmail.com (M.H.A.); kincses.annamaria@med.u-szeged.hu (A.K.); spengler.gabriella@med.u-szeged.hu (G.S.); 2Department of Microbiology and Biotechnology, Faculty of Food Science, Szent István University, 1118 Budapest, Hungary; ahmad.mouwakeh@hotmail.com (A.M.); farkas.csilla@etk.szie.hu (C.M.-F.); kisko.gabriella@etk.szie.hu (G.K.)

**Keywords:** *Chlamydia trachomatis*, antibacterial activity, *Nigella sativa*

## Abstract

Urogenital tract infection caused by obligate intracellular bacterium *Chlamydia trachomatis* D (*Ctr*D) is a leading cause of sexually transmitted diseases. Essential oil (EO) of *Nigella sativa* has a broad antimicrobial spectrum. The aim of this study was to evaluate the antimicrobial activity of the bioactive compounds (p-cymene, thymoquinone, carvacrol, and thymol) of *N. sativa* EO against *Ctr*D. The cytotoxic effects of the compounds were determined by MTT assay. In order to quantify the anti-chlamydial activity of the compounds, HeLa cells were infected with *Ctr*D or *Ctr*D treated previously with the compounds. The titer of the infectious *Ctr*D was determined by indirect immunofluorescence assay. The minimum inhibitory concentrations of the compounds were evaluated by direct quantitative PCR. None of the compounds showed a cytotoxic effect on HeLa cells in the concentrations tested. According to the immunofluorescence assay, all of the compounds significantly inhibited the growth of *Ctr*D. The quantitative PCR revealed that the minimum concentration that exerted anti-chlamydial activity was 3.12 µM in the case of thymoquinone and p-cymene, while that of carvacrol and thymol was 6.25 µM. Therefore, it can be concluded that bioactive compounds of *N. sativa* EO could be used as effective antimicrobial agents against *Ctr*D.

## 1. Introduction

Although antibiotic therapy eliminates bacterial infections, there is emerging evidence of bacteria developing antimicrobial resistance (AMR). AMR together with a lack of development of new antimicrobial agents has become a global public health concern [1]. 

*Chlamydia trachomatis* is an obligate intracellular bacterium that causes a wide spectrum of human diseases, such as genitourinary, pulmonary, and ocular infections. The most common genitourinary infections caused by *C. trachomatis* serovars D to K are mucopurulent cervicitis in females and non-gonococcal urethritis in males. Additionally, cases of untreated infections can lead to various complications, such as pelvic inflammatory disease (PID), ectopic pregnancy, chronic prostatitis, and infertility [2]. *Chlamydia* spp. are characterized by typical lifecycles. First, the elementary body (EB), which is the infectious form, infects the host cell. After the EB enters the host cell, the formation of inclusion occurs and the EB transforms into the reticulate body (RB). The RB is characterized by its high metabolic activity and further division by binary fission. This process subsequently results in the filling of the entire cytoplasm and dislocation of the nucleus. Approximately 24 to 72 h later, there is a final transition of RBs into EBs that ends with host cell lysis [3]. 

Chlamydial infections can be managed by azithromycin, tetracyclines, and fluoroquinolones. However, rates of clinical treatment failures range from 5% to 23%, which might be attributed to AMR [4]. Azithromycin resistance of *C. trachomatis* serovar L2 is caused by a mutation in the *rplD* gene that codes for ribosomal protein L4. This alteration results in a declining activity of antibiotics by interfering with protein synthesis [5]. *C. trachomatis* resistance to fluoroquinolone is attributed to a point mutation of the *gyrA* [6]. Although chlamydiae are replicating in a membrane bound vacuole, horizontal gene transfer could be involved in the occurrence of AMR. A recent study reported that tetracycline resistance in *Chlamydia* spp. is associated with the horizontal gene transfer of antibiotic resistance genes (*tetC*, *tetR*), which encode efflux pumps [7]. AMR of chlamydiae could be the result of selective pressure of continuous exposure to antimicrobial drugs at subinhibitory concentrations [4]. Furthermore, chlamydiae can transform to persistent forms, which further enhances their resistance to antimicrobial drugs [8]. 

Phythochemicals have garnered attention over the past decade because of their therapeutic potential against a wide range of pathogenic microorganisms. The antimicrobial activity of essential oils (EOs) extracted from medicinal plants is well demonstrated [9,10]. EO obtained from *Nigella sativa* (black cumin), which is rich in phenolic compounds, has a broad antimicrobial spectrum including both Gram-negative and Gram-positive bacteria, viruses, parasites, and fungi [11]. In addition, *N. sativa* EO effectively reduced the development of bacterial biofilm of *Staphylococcus aureus* in an in vitro study [12]. Among the phenolic constituents, p-cymene (p-cy) and thymoquinone (Thq) are the major components of *N. sativa* EO [13]. Carvacrol (Car) and thymol (Thy) can also be found in the EO extracted from *N. sativa* [14,15]. 

To the best of our knowledge, only one study has been published in association with the anti-chlamydial activity of EOs or other formulations of phythochemicals. Specifically, the anti-chlamydial effect of EO obtained from *Mentha suaveolens* was investigated on the lymphogranuloma venereum strain of *C. trachomatis* [16]. The aim of our study was to evaluate the antimicrobial activity of *N. sativa* EO and its bioactive compounds (p-cy, Thq, Car, and Thy) against *C. trachomatis* serovar D.

## 2. Materials and Methods 

### 2.1. Bacterial Strain and Cell Line

*Chlamydia trachomatis* (serovar D, UW-3/Cx) was propagated on HeLa 229 cells (ATCC, CCL-2.1). The infected cells were purified by density gradient centrifugation, as previously described [17]. The titer of infectious elementary bodies (EBs) was determined by indirect immunofluorescence assay and was expressed in inclusion forming unit/mL (IFU/mL) [18]. HeLa cells were maintained in minimal essential medium (MEM) comprising 10% fetal bovine serum, 2 mM L-glutamine, 1 × nonessential amino acids, 1 × MEM vitamins, 25 μg/mL gentamicin, and 1 μg/mL fungizone [19].

### 2.2. Essential Oil and Active Compounds

*N. sativa* EO extraction was performed as reported earlier [14]. Thymoquinone (Thq), thymol (Thy), and carvacrol (Car) were purchased from MilliporeSigma ( St. Louis, MO, USA) and p-cymene (p-cy) was purchased from Alfa Aesar (Haverhill, MA, USA). EO, Thy, and Thq were diluted using dimethyl sulfoxide (DMSO, MilliporeSigma), while ethanol was used as diluent for Car and p-cy to prepare stock solutions, and further dilutions were performed with medium used for the maintenance of HeLa cells.

### 2.3. Cytotoxicity Assay

The effects of increasing concentrations of the compounds on HeLa cell growth were tested as described by Żesławska et al. [20]. Briefly, 2 × 10^4^ HeLa cells in 100 μL of medium were added to each well, with the exception of the medium control wells. After an overnight incubation period, the compounds were diluted and added to the cells. Initial concentrations of the bioactive compounds were 100 μM, while in the case of the EO it was 0.04% (*v*/*v*). After 48 h, 20 μL of MTT (thiazolyl blue tetrazolium bromide, MilliporeSigma) solution (from a 5 mg/mL stock) were added to each well. After 4 h, 100 μL of sodium dodecyl sulfate (SDS, MilliporeSigma) was added to each well and the plates were further incubated at 37 °C overnight. The cell growth was determined by measuring the optical density. Inhibitory concentration 50 (IC_50_) was evaluated, where the compounds reduced the growth of the treated HeLa cells by 50%.

### 2.4. Anti-Chlamydial Assay

EBs of *C. trachomatis* D (4 × 10^4^ IFU/mL) were incubated with *N. sativa* EO (0.0025% *v*/*v*) and its bioactive compounds at various concentrations (25, 50 μM) in a sucrose–phosphate–glutamic acid buffer (SPG) for 2 h at 37 °C. As a control, *C. trachomatis* D was also incubated in SPG alone. To quantify the anti-chlamydial effects of compounds, confluent HeLa cells were infected with compounds-treated *C. trachomatis* D or the non-treated controls. After 48 h, the cells were fixed with acetone at −20 °C for 10 min, and the number of *C. trachomatis* D inclusions was determined by immunofluorescence assay [18]. 

### 2.5. Determination of Minimal Inhibitory Concentrations

Minimal inhibitory concentrations (MICs) of the effective compounds were evaluated by a previously described method [21]. Briefly, HeLa cells were infected with *C. trachomatis* D (1 multiplicity of infection) and treated with the compounds in two-fold dilutions for 1 h at 37 °C. The initial concentrations of compounds were 100 μM. HeLa cells infected with *C. trachomatis* D alone were used as controls. After 48 h, the cells were washed and resuspended in water. The number of infectious EBs was determined by direct quantitative PCR using the following primers: *pykF* forward 5’-GTT GCC AAC GCC ATT TAC GAT GG-3’; *pykF* reverse 5’-TGC ATG TAC AGG ATG GGC TCC TA-3’.

### 2.6. Statistical Analysis

All values are expressed as a mean ± standard deviation of three replicates from three independent experiments. Statistical analysis of the data was carried out with SigmaPlot for Windows Version 12.0 software (Systat Software, San Jose, CA USA), using the two-tailed *t*-test for independent samples. Differences were considered statistically significant at *p* < 0.05.

## 3. Results

### 3.1. Cytotoxicity Assay

Before the assessment of the anti-chlamydial activity of the compounds, HeLa cells were incubated with increasing concentrations of *N. sativa* EO and its bioactive components for 48 h. The cell viability was measured by MTT assay, and IC_50_ values were evaluated (Table 1). No significant cytotoxicity was observed following the exposure of HeLa cells to p-cy, Thq, Car, and Thy up to 100 μM. By contrast, *N. sativa* EO exerted cytotoxic properties towards HeLa cells and its IC_50_ value was defined at 0.009% (*v*/*v*). A four-fold lower concentration than its IC_50_ was used in the anti-chlamydial assay, in order to avoid the direct toxic effects of EO. 

### 3.2. Anti-Chlamydial Assay

In order to determine the anti-chlamydial activity of *N. sativa* EO and its compounds, 0.0025% (*v*/*v*) of EO were incubated with the EB suspension for 2 h. The active components of EO were tested at concentrations of 25 or 50 μM. As shown in Figure 1, all of the compounds tested significantly reduced the infectivity yield after 2 h of treatment. Treatment of EBs with *N. sativa* EO completely inhibited the replication of *C. trachomatis* D. The same results were observed when the chlamydial EB suspension was treated with Thq, Car, or Thy at concentrations of 50 μM (Figure 2). Moreover, exposure to 25 μM of Thq was able to reduce the formation of inclusions by 100%. Among the components of *N. sativa*, p-cy proved to be the least effective, although it inhibited the growth of *C. trachomatis* D by more than 50% even at the lowest concentration examined. 

### 3.3. Determination of Minimal Inhibitory Concentrations

As all of the bioactive compounds tested showed antimicrobial activity in the anti-chlamydial assay, their MICs were evaluated by direct quantitative PCR (Figure 3). HeLa cells were infected with *C. trachomatis* D and at the same time treated with two-fold serial dilutions of p-cy, Thq, Car, or Thy. After 1 h, the cells were washed and the medium was replaced. Direct PCR was performed from the cells 48 h later in order to determine the number of infectious *C. trachomatis* D. Untreated but infected cells were used as controls. Treatment of the EBs with 100 μM of Thq for 1 h completely inactivated the EBs of *C. trachomatis* D. We did not observe complete inhibition for the other bioactive compounds—even at the highest concentrations tested. The MICs of p-cy and Thq were defined at 3.12 μM, while the lowest concentration that significantly inhibited the replication of *C. trachomatis* D was 6.25 μM in the cases of Car and Thy. 

## 4. Discussion

The emergence of AMR is considered as a major public health problem due to the appearance of reduced or missing response of microorganisms to the applied antimicrobial agents. *C. trachomatis* infection is the most commonly reported sexually transmitted, bacterial infection, with an estimated 131 million new cases [22]. In addition, it has been found that *Chlamydia* spp. possess several different mechanisms associated with AMR development, despite their unique lifecycle characteristics. Under exposure to certain conditions, such as the presence of interferon-γ, β-lactam antibiotics, or deprivation of nutrients, *C. trachomatis* can transform to a persistent state, which can be defined by reduced replication and the occurrence of aberrant bodies [8]. Moreover, a recently published study demonstrated that azithromycin, which is the first choice drug in the therapy of chlamydial infections, could induce persistent infection in vitro [23]. Subinhibitory concentrations of the antimicrobial drugs were also able to induce AMR of certain chlamydial strains [5,6,7]. The ideal anti-chlamydial agents would be able to inhibit the growth of chlamydiae without exerting selective pressure for the development of AMR. The main advantages of natural-based products are that they apply less selective pressure against pathogens and exert remarkable effects on the inhibition of efflux pumps and AMR reversal [24,25]. The most common natural bioactive agents are volatile phenolic compounds, such as p-cy, Thq, Car, Thy, cinnamaldehyde, eugenol, limonene, and menthol, which are secondary metabolites of medicinal plants [15]. 

Our previous study revealed that EO extracted from *N. sativa* inhibited the growth of *S. aureus*, including methicillin resistant *S. aureus,* and exerted antibiofilm activity. Regarding the bioactive compounds of *N. sativa* EO, both staphylococcus strains were sensitive to Thq and Car [12]. In this present study, we demonstrated that the *N. sativa* EO was able to completely inactivate the EBs of *C. trachomatis* D after 2 h exposure time at a concentration that was four-fold lower than its IC_50_ evaluated on HeLa cells. Moreover, all of the bioactive constituents (p-cy, Thq, Car, Thy) showed a direct antibacterial effect against *C. trachomatis* D. As only one study related to anti-chlamydial activity of EOs has been published, further studies are needed to clarify the exact mechanisms of their effects. Car and Thq were able to damage the cell membrane of *S. aureus* and *Listeria monocytogenes* [14,26]. Thy, which is the most common constituent of EOs obtained from *Thymus* spp. and p-cy, exerted antimicrobial activity against a broad spectrum of pathogens, including Gram-positive and Gram-negative bacteria and fungi. Similar to other monoterpenes, Thy and p-cy were able to damage bacterial lipid membranes; therefore, the possible mechanisms related to anti-chlamydial activity of Thq, Car, Thy, and p-cy might be associated with the disruption of the lipid bilayers [15,27]. 

MICs of the compounds were evaluated by direct quantitative PCR and defined at 6.25 (Car, Thy) and 3.12 μM (p-cy, Thq), respectively. We were not able to detect complete inhibition of *C. trachomatis* D, except in the case of Thq, which could be the result of the shorter exposure time (1 h). This finding supports the fact that the efficacy of their antimicrobial activity is time-dependent [16]. 

We are planning further experiments to evaluate the antimicrobial effects of *N. sativa* EO and its bioactive compounds on intracellularly replicating *C. trachomatis* RBs and their synergistic effects with clinically used antibiotics. 

## 5. Conclusions

It can be concluded that bioactive compounds of *N. sativa* EO inhibited the replication of *C. trachomatis* D in vitro. These findings suggest that *N. sativa* EO or its bioactive constituents could be used as effective antimicrobial agents against *C. trachomatis* D. As numerous EOs possess antimicrobial activity and in turn can enhance the effect of antibiotics, further studies could support the use of bioactive components of *N. sativa* EO as potential phytotherapeutics in anti-chlamydial therapy.

## Figures and Tables

**Figure 1 microorganisms-07-00370-f001:**
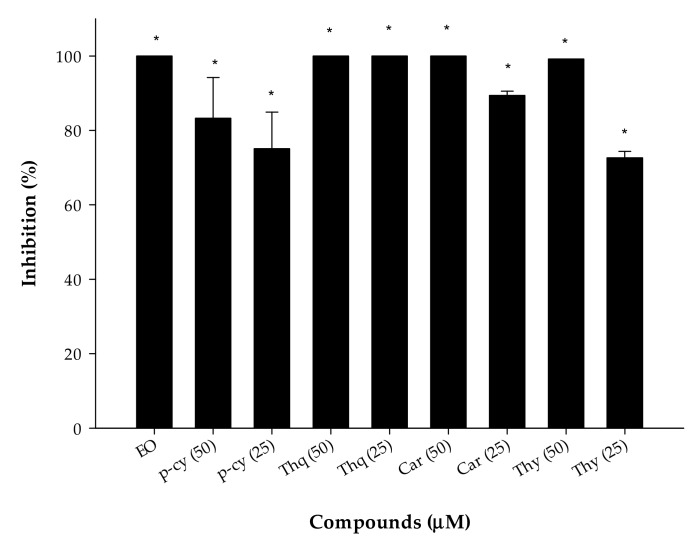
Anti-chlamydial effects of compounds at 25 and 50 μM. The *N. sativa* essential oil (EO) was tested at a concentration of 0.0025% (*v*/*v*). p-cy: p-cymene; Thq: thymoquinone; Car: carvacrol; Thy: thymol; * *p* < 0.05.

**Figure 2 microorganisms-07-00370-f002:**
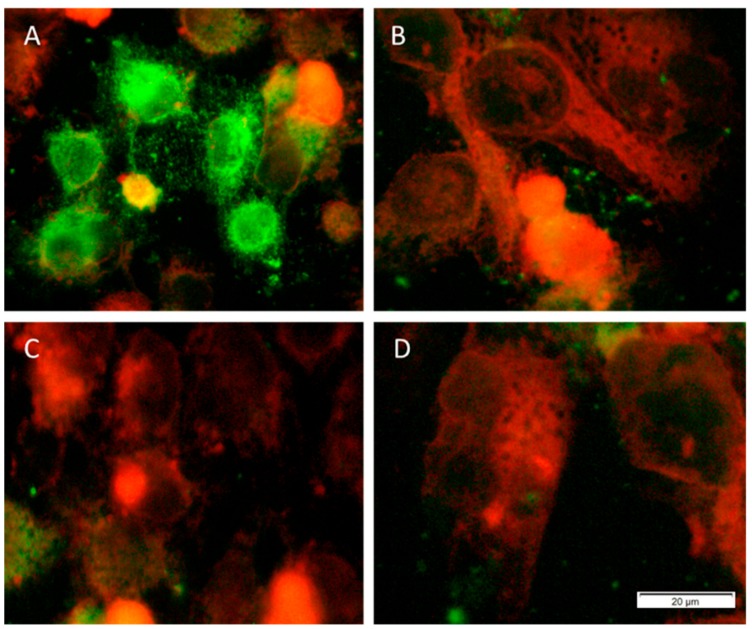
Immunofluorescence-stained inclusions of *C. trachomatis* D in HeLa cells. The cells were infected with (**A**) *C. trachomatis* D alone or with *C. trachomatis* D pre-incubated with (**B**) thymoquinone; (**C**) carvacrol; (**D**) thymol at a concentration of 50 μM. Pictures were acquired by a digital camera attached to a fluorescence microscope.

**Figure 3 microorganisms-07-00370-f003:**
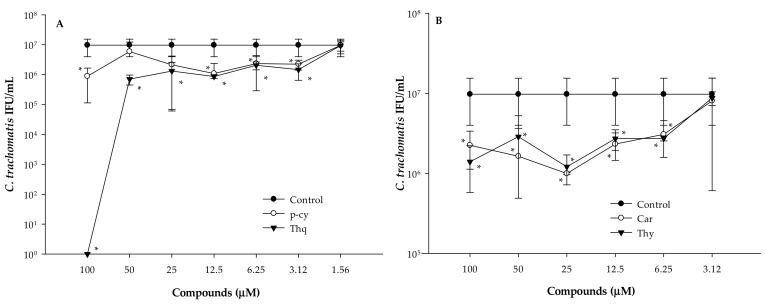
Inhibitory effects of the bioactive compounds of *N. sativa* EO on *C. trachomatis* D at different concentrations evaluated by direct quantitative PCR. HeLa cells infected with *C. trachomatis* D alone were used as controls. (**A**) p-cy: p-cymene; Thq: thymoquinone; (**B**) Car: carvacrol; Thy: thymol; * *p* < 0.05.

**Table 1 microorganisms-07-00370-t001:** Cytotoxic effects of *Nigella sativa* essential oil (EO) and its bioactive compounds on HeLa cells.

Compounds	IC_50_
p-cymene	>100 μM
thymoquinone	>100 μM
carvacrol	>100 μM
thymol	>100 μM
*N. sativa* essential oil	0.009% (*v*/*v*)

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
