# Peer review of "Bioactive Compounds of Nigella Sativa Essential Oil as Antibacterial Agents against Chlamydia Trachomatis D"

_microorganisms, 2019, doi:10.3390/microorganisms7090370_

Round 1

Reviewer 1 Report

The authors of this study have presented the antibactericidal effects of the bioactive compounds of Nigella Sativa essential oil. However, the novelty or scientific quality of this study has to be improved by performing following experiments:

Every finding should be demonstrated quantitatively as well as qualitatively. Hence, microscopic images of significant reduction in Chlamydial infection should be included in the results. Authors have pre-incubated EBs with bioactive compounds of NS. However, it would also be more appropriate for the authors to show the therapeutic effects of these compounds by adding the compounds to the cells already infected with Chlamydia (2hrs, 4hrs, 12hrs and 24hrs post infection) and looking at their inhibitory effects.

These above-mentioned experiments will improve the rigor and novelty of the work.

Reviewer 2 Report

This brief communication reports the effects against C. trachomatis of the essential oil of N. sativa, and of some its components (p-cymene, thymoquinone, carvacrol and thymol), previously shown active against S. aureus.

All the assayed bioactive compounds showed antichlamydial activity, without significant effects on HeLa cells. However, only thymoquinone was able to completely inactivate the elementary bodies of C. trachomatis, while all the compounds showed significant inhibition of the replication in the low micromolar range.

Overall, the manuscript is well presented, the experiments are well conduced and the results support the conclusions. Although very preliminary, this results are encouraging and pave the way for the development of novel antichlamidial compounds.

Round 2

Reviewer 1 Report

Accept it as it is.